# The Study of the Influence of ZrO_2_ Precursor Type and the Temperature of Annealing on the Crystallization of the Tetragonal Polymorph of ZrO_2_ in Zirconia-Silica Gels

**DOI:** 10.3390/gels8110724

**Published:** 2022-11-09

**Authors:** Anna Adamczyk

**Affiliations:** Faculty of Materials Science and Ceramics, AGH University of Science and Technology, Al. Mickiewicza 30, 30-059 Kraków, Poland; aadamcz@agh.edu.pl

**Keywords:** ZrO_2_-SiO_2_ system, sol-gel method, FTIR spectroscopy, X-ray diffraction, SEM, *t*-ZrO_2_

## Abstract

Materials of the ZrO_2_-SiO_2_ system were obtained by the sol-gel method applying two different types of ZrO_2_ precursors: zirconium (IV) n-propoxide Zr(OC_3_H_7_)_4_ and zirconium (IV) acetate Zr(OOC_2_H_3_)_4_ (organic acetic acid salt) while commonly used tetraethoxysilane TEOS was selected as SiO_2_ introducing one. ZrO_2_ concentration in synthesized samples varied from 20% to 50% (mol.). After drying for 28 days, all gels were annealed at 500 °C, 1000 °C, and 1200 °C in air. FTIR spectroscopy together with XRD diffraction was selected as the two main structure research methods. SEM microscopy was applied to analyze the local chemical compositions of samples and to observe the morphology of gels’ surfaces. The analysis of FTIR spectra and XRD diffraction patterns allowed us to recognize different ZrO_2_ polymorphs which appeared in the samples depending strongly as well on ZrO_2_ precursor type as on the temperature of annealing. Samples synthesized by using the zirconium (IV) n-propoxide contained both cubic and tetragonal zirconia phases in general but showed the tendency of the increasing *t*-ZrO_2_ content in gels richer in ZrO_2_ and heated up to 1200 °C. However, in materials obtained applying zirconium (IV) acetate, the first detected at 500 °C phase was *t*-ZrO_2_ which was then conversing to *m*-ZrO_2_ form with the increasing temperature in case of samples rich in ZrO_2_. Meanwhile, *t*-ZrO_2_ was the predominant phase in samples of the lower content of ZrO_2_ but annealed at higher temperatures. By the analysis of changes in band profiles and positions, one can draw conclusions that the structure of studied samples is mostly built up of an amorphous silica matrix, in which different types of zirconia polymorphs create their own crystal lattice. The presence of the particular polymorph depends strongly on the type of zirconia precursor and the temperature of annealing.

## 1. Introduction

The sol-gel method is one of the most often used methods of synthesis of homogeneous and nanoscaled materials. Moreover, this method allows for the synthesis of samples at much lower temperatures but without wasting their precious properties, especially those connected with nanoparticles size. The structure of gels of higher silica content can be described as built of silica matrix with different additives (e.g., oxides) incorporated in it. This fact has to be reflected in the properties of obtained materials. Silica as well as materials from the ZrO_2_-SiO_2_ system have got high thermal, electrical, and magnetic conductivity [1]. They also present very good optical properties [2]. Amorphous gels of the ZrO_2_-SiO_2_ system usually exhibit a high reflection coefficient and simultaneously good mechanical and thermal properties [3]. 

One of the problems observed in the ZrO_2_-SiO_2_ system is connected with the synthesis and the stabilization of metastable, tetragonal zirconia polymorph. This tetragonal *t*-ZrO_2_ is valuable because of its advantageous properties towards other ZrO_2_ forms. ZrO_2_ crystallizes in three main forms: *m*-ZrO_2_ (monoclinic zirconia) crystallizing up to 1170 °C, *t*-ZrO_2_ (tetragonal one) whose presence can be confirmed between 1170 °C and 2300 °C and as third one, *c*-ZrO_2_ (of a cubic structure), stable from 2300 °C up to the melting point at 2680 °C. But this important one is the metastable tetragonal ZrO_2_ which is very often observed in samples heated in the range of 700–1000 °C. This tetragonal ZrO_2_ polymorph can easily transform into the monoclinic one because of e.g., the influence of temperature or pressure changes [4]. The metastable *t*-ZrO_2_ can be stabilized by three main factors: compressive stress, stoichiometry level, and by crystallite size [5,6].

The stabilization of the ZrO_2_ tetragonal phase can be achieved by the introduction of MgO or Y_2_O_3_ or other alkalinoterrous oxides into the structure of zirconia-containing systems. This procedure allows to obtain of FSZ (Fully Stabilized ZrO_2_), PSZ (Partially Stabilized ZrO_2_), or TPZ (Tetragonal zirconia Polycrystals) phases [7]. Stress which can appear in the structure of annealed zirconia-containing materials e.g., during the oxidation processes can also influence the obtaining of stable *t*-ZrO_2_ [8]. The commonly used sol-gel method enables to stabilize and synthesize tetragonal zirconia (even in form of nanoparticles) by the use of different alkoxides or different synthesis parameters such as, e.g., type of solvent, temperature, pH, pressure, etc. [9,10,11]. However one can find works where authors did not observe the correlation between the presence of t-ZrO_2_ and Zr precursor type. Such a situation was observed in the case of applying three zirconia precursors: a zirconium oxychloride, a zirconium oxynitrate, and a zirconium acetate where their lack of influence on the gels structure and the crystallization of tetragonal ZrO_2_ was described in [12]. Besides the studies of the zirconia-silica system suggest that ZrO_2_ incorporated into the silicate matrix can influence the formation of amorphous or poorly crystallized materials [13,14,15].

In this work, the sol-gel method was selected for obtaining gels of high silica content with the addition of different amounts of ZrO_2_. During the synthesis two compounds, Zr (IV) n-propoxide and Zr (IV) acetate (an alkoxide and an organic acid salt respectively) were selected as zirconia precursors so one has got the possibility of the comparison of the influence of these two different type compounds on the structure of synthesized gels. The usage of different types of ZrO_2_ precursors should also affect some selected properties of samples of this system. 

Thus, the main target of this work can be described as the structure research of the synthesized materials in relation to the ZrO_2_ precursor type used during the synthesis. The focus was put on the simplest way of obtaining the stabilized tetragonal form of zirconia. This second aim could be realized by applying the different types of ZrO_2_ precursors and the proper temperature of annealing.

## 2. Results and Discussion

Samples were synthesized in two series signed as samples of series T (for samples synthesized applying zirconium (IV) n-propoxide) and of series C—samples obtained using zirconium (IV) acetate. For both series, samples composition were selected according to ZrO_2_:SiO_2_ ratio as 1:1, 1:2, 1:3 and finally 1:4. In both series, samples were also named according to silica content e.g., sample 3T means gel of composition given by ZrO_2_:SiO_2_ ratio as 1:3. The synthesis of samples of both series and the selected structure research methods are described in details in Section 4.

### 2.1. X-ray Diffraction

X-Ray diffraction patterns of all dried and annealed samples of series T and C as well as of pure ZrO_2_ annealed gel are shown in Figure 1, Figure 2 and Figure 3.

Measurements of pure, dried and annealed zirconia gel (Figure 1) gave a lot of information on the structural changes caused by the temperature treatment from 500 °C to 1200 °C. The first reflections assigned to the cubic and the tetragonal forms of ZrO_2_ were observed at 500 °C. With the increasing temperature, the polymorphic transformation was observed and as a result another two phases: the monoclinic *m*-ZrO_2_ together with *t*-ZrO_2_ were identified at 1000 °C whereas only one *m*-ZrO_2_ with very little addition of *t*-ZrO_2_ predominated at 1200 °C.

All annealed samples of T series remained amorphous after annealing at 500 °C whereas the first traces of crystallization of ZrO_2_ could be recognized at 1000 °C (Figure 2). XRD phase analysis showed evident differences and changes in the structure of gels heated to 1000 °C and then up to 1200 °C. In the case of samples 1T and 2T, richer in ZrO_2_, phase analysis allowed to identify tetragonal and cubic forms of ZrO_2_ at 1000 °C while at 1200 °C only one tetragonal *t*-ZrO_2_ was observed, accompanied by very little addition of impossible to specify the form of silica [16,17].

For comparison, diffraction patterns of rich in silica samples 3T and 4T pointed to the presence of the same phases *t*-ZrO_2_ and *c*-ZrO_2_ at 1000 °C as in the case of samples 1T and 2T but at 1200 °C both phases were present only in sample 3T together with a very small addition of unknown silica. The diffraction pattern of sample 4T heated at 1200 °C showed reflections due to the single dominant phase—the cubic *c*-ZrO_2_ together with a few very weak reflections due to unidentified SiO_2_. 

Samples of the C series behaved in quite a different way (Figure 3). First of all, they all started to crystallize at 500 °C and their diffraction patterns at this temperature exhibited the presence of reflections assigned only to the tetragonal form of ZrO_2_. Peaks characteristic of *t*-ZrO_2_ could be found in the diffraction patterns of all samples at each temperature of annealing but they predominated in the XRD patterns of samples 3C and 4C as the single identified phase (with a very little amount of silica crystalline phase) at 1000 °C and 1200 °C. In the diffraction patterns of samples 1C and 2C annealed at 1000 °C, reflections of both, *t*-ZrO_2_ and *m*-ZrO_2_ were identified but at 1200 °C, one phase was predominating—the monoclinic *m*-ZrO_2_ [18,19,20].

### 2.2. FTIR Spectroscopy

In the FTIR spectra of all dried gels of pure ZrO_2_ and gels of series T and C corresponding bands are observed at about 3500 cm^−1^ and at 1650 cm^−1^ assigned to the vibrations of OH^−^ groups and the vibrations in the molecular water particles, respectively. In Figure 4, Figure 5 and Figure 6, one can distinguish the band at 1650 cm^−1^ up to 500 °C. These bands reduced their intensity and finally vanished with the rise of temperature. Besides, there were another characteristic, intensive bands that appeared in the spectra of all dried gels, at about 1567 cm^−1^ and 1455 cm^−1^, which can be due to Zr–O–C and Zr–OH bond vibrations, respectively. The mentioned bands disappeared with the increase in the annealing temperature which is caused by the evaporation of water and the decomposition of organic units during annealing. This fact confirmed the suggested assignments of these bands to the mentioned Zr–O–C and Zr–OH linkages.

In the IR spectra of zirconia-silica annealed gels, bands assigned to the vibrations of different Si–O bonds can be distinguished. The most intensive band at about 1085–1095 cm^−1^ can be due to the asymmetric stretching vibration of the Si–O bond, while the weaker band at 788–805 cm^−1^ can be assigned to the symmetric stretching vibrations of such linkage [21].

Two additional bands, at about 1220–1240 cm^−1^ and at 940 cm^−1^ (observed in some spectra), connected with the double Si=O bond vibrations and with the vibrations of Si–O^−^ broken bridges, respectively, can also be distinguished. 

Another group of bands that can be assigned to the bending vibrations of Si–O bonds can be found in the range of 400–700 cm^−1^ [22]. The bands at about 450–470 cm^−1^ are, according to [23], due to the bending vibrations of O–Si–O bridges while bands in the range of 560–650 cm^−1^ can be assigned to the pseudolattice vibrations connected with the presence of structure built up of silica rings. But it is important and worth mentioning that in the same range bands due to the stretching vibrations of Zr–O bonds can also be found [24,25,26,27]. The assignment of bands at 420 cm^−1^, 580 cm^−1^ and 650 cm^−1^ to the stretching vibrations of Zr-O bonds is more probable due to the progressive crystallization of different forms of zirconia in these samples, which agrees with the X-ray analysis results. Furthermore, the band at about 640–650 cm^−1^ can be found in the spectra of tetragonal ZrO_2_ [7].

In the spectra of T series samples (Figure 5), the intensity of bands at 800 cm^−1^ and 1100 cm^−1^ increases which suggests the progressive polymerization in the silica lattice. At the same time, the intensity of the band at 1220 cm^−1^ also increases together with the progressive crystallization of different zirconia polymorphs (confirmed by XRD results). This fact is also confirmed by the appearance and the increasing intensity of the band at about 590 cm^−1^, which can be connected with the crystallization of *t*-ZrO_2_. 

In the spectra of C series gels (Figure 6), the intensity of bands at 800 cm^−1^, 1100 cm^−1,^ and 1220 cm^−1^ also increases what is probably connected with the increasing content of silica in samples and simultaneously the increasing concentration of Si–O bonds in the structure. In the spectra of all 1C, 2C, 3C, and 4C samples a few weaker bands at 420–460 cm^−1^ and around 580 cm^−1^ can be distinguished. Those bands can also be assigned to Zr–O connections but not only in the tetragonal form of ZrO_2_. The assignment of these bands to Zr–O vibrations in a proper form of zirconia depends on the result of XRD phase analysis.

Summarizing, according to the IR spectra (together with the results of the XRD analysis) there is no possibility of the existence of a continuous, periodic silica network in the studied zirconia-silica samples. Probably, the structure of samples is built up of silica matrix (amorphous or partially crystallized) with different forms of zirconia polymorphs creating their own crystal lattice, in dependence on the type of applied zirconia precursor and the temperature of annealing. Such type of structure of obtained zirconia-silica materials was also observed by other researchers [13,14,15].

### 2.3. Scanning Electron Microscopy (SEM)

All SEM images of the surface of zirconia-silica gels heated up to 1200 °C (Figure 7) look similar, independently of the type of ZrO_2_ precursor used. The only exception is 1T gel annealed up to 1200 °C (Figure 7a) which exhibits the presence of crystallites of spherical shape rich in Zr atoms (according to EDS results—Figure 8) but simultaneously containing also significant amounts of silicon and oxygen atoms. The same type of spherical objects were observed on the surface of pure ZrO_2_ gel synthesized using zirconium (IV) n-propoxide and annealed up to 1200 °C (data not presented in this work). The XRD measurement confirmed the presence of tetragonal zirconia in these samples. Similar crystallites were also reported in zirconia-silica samples [10] annealed at 1200 °C during the transformation from tetragonal to monoclinic ZrO_2_. Thus, one can attribute the spherical objects to crystallites of tetragonal zirconia whose presence in this sample is also confirmed by XRD measurements. Such objects are not identified on the surface of the remained gels, even those for which XRD phase analysis confirmed the presence of the tetragonal ZrO_2_ polymorph in their structure.

Although the surface of the studied samples look very similar (with this mentioned exception of 1T gel), analyzing EDS spectra one can observe the disturbances of the relation of Zr:Si atoms concentration in gels of series C synthesized with Zr (IV) acetate (Figure 9b,c) with respect to the planned compositions of samples. It might be connected with the local fluctuation of both Zr and Si atoms concentration in samples which may result from the synthesis conditions and also might influence on a faster crystallization of zirconia polymorphs during the annealing of samples.

## 3. Conclusions

Zirconium (IV) n-propoxide, zirconium (IV) acetate, and TEOS (tetraethylorthosilane) were applied as ZrO_2_ and SiO_2_ precursors. All samples were dried and then annealed from 500 °C to 1200 °C in air.

X-ray phase analysis allowed us to observe that samples synthesized applying zirconium n-propoxide (series T) were amorphous below 1000 °C while those obtained using zirconium acetate (series C) began to crystallize at 500 °C.

In samples of series T, the first observed phases were tetragonal *t*-ZrO_2_ and cubic *c*-ZrO_2_ in samples 1T and 2T. Those phases converted into one phase *t*-ZrO_2_ at 1200 °C in both samples. In samples, 3T and 4T rich in silica, the mixture of two phases *t*-ZrO_2_ and *c*-ZrO_2_ was detected in the 3T sample but only one phase, cubic *c*-ZrO_2_ was observed in the sample 4T at 1200 °C. 

The first phase detected in samples of series C synthesized applying zirconium acetate, was tetragonal *t*-ZrO_2_, which conversed into monoclinic *m*-ZrO_2_ with the increasing temperature in sample 1C and 2C (rich in ZrO_2_) but remained as the one predominant phase in samples 3C and 4C of smaller content of ZrO_2_, despite the rise of temperature.

In general, the structure of the annealed samples probably consists of the amorphous silica matrix, in which different forms of ZrO2 are crystallizing in dependence on the type of ZrO_2_ precursor and the temperature of annealing.

SEM studies allowed us to observe the spherical crystallites of probably tetragonal polymorphs of ZrO_2_ in samples of series T of higher ZrO_2_ content. Such crystallites were not observed in samples of series C, prepared using zirconium acetate. As it was mentioned the presence of these spherical objects pointed to the crystallization of *t*-ZrO_2_ in samples. 

Applying both zirconia precursors, one is able to obtain one predominant phase of tetragonal zirconia in the annealed gels. The presence of *t*-ZrO_2_ is strongly connected with the Zr precursor type, the molar ratio ZrO_2_:SiO_2_ in samples, and the temperature of annealing.

## 4. Materials and Methods

All samples were synthesized by the sol-gel method, using two different type of zirconia precursors: zirconium (IV) n-propoxide Zr(OC_3_H_7_)_4_ (70% solution in 1-n propanol, Sigma-Aldrich, St. Louis, MO, USA) and zirconium (IV) acetate Zr(OOC_2_H_3_)_4_ (zirconium acetate solution in dilute acetic acid, Sigma-Aldrich, St. Louis, MO, USA). As a SiO_2_ precursor, commonly known as TEOS (tetraethoxysilane Si(OC_2_H_5_)_4_ (Fluka 98%, St. Gallen, Switzerland) was selected. During the synthesis, 98% or 96% ethanol was applied as the solvent however in some synthesis ethanol was used as the mixture with distilled water at the selected ethanol:water ratio.

As the first one, 5% weight silica sol was prepared. To obtain the proper concentration of silica, TEOS was dissolved in 98% ethanol according to the molar ratio 0.085:0.085 respectively. The second solution of HCl (35% weight), 98% ethanol and redistilled water (in the molar relation 0.005:0.844:0.340) were then prepared. After stirring both solutions separately for 10 min, the second solution was dropped very slowly to the first preparation, after that the obtained sol was mixed for the next 2 h and subjected to aging [23]. Then, 4% weight ZrO_2_ sol was prepared to apply zirconium (IV) n-propoxide [28]. In the beginning, zirconium (IV) n-propoxide was dissolved in 96% ethanol with the molar ratio Zr (IV) n-propoxide to ethanol as 0.025:0.070. Then ethanol, acetic acid, and a last portion of ethanol were dropped slowly in stages, maintaining the appropriate molar ratio of the components 0.105:0.141:0.703. After that, the solution of ethanol and distilled water (0.176:0.388) was added to the sol which was then stirred for the next 1.5 h.

As was mentioned in chapter 2, four compositions of samples corresponding to the molar ratio of ZrO_2_:SiO_2_ as 1:1, 1:2, 1:3, and finally 1:4 were selected for both precursors. All ZrO_2_-SiO_2_ gels synthesized with zirconium (IV) n-propoxide were obtained by mixing in a proper ratio, two one-component 4% zirconia and 5% silica sols while gels obtained using zirconium acetate were prepared from the beginning as two-component sols of the selected compositions [29]. To prepare a sol containing ZrO_2_ and SiO_2_ in a 1:1 ratio, an ethanol, TEOS, and a nitric acid were mixed according to the molar ratio 0.162:0.009:0.049. After 30 min of stirring zirconium acetate was dropped slowly applying the molar ratio Zr(OOC_2_H_3_)_4_:TEOS = 0.065:0.009. Then a whole preparation was homogenized for a further 90 min. Applying the same procedure and the proper molar relations, sols of 1:2, 1:3, and 1:4 compositions were also synthesized. 

All samples synthesized with zirconium (IV) n-propoxide were then called T series and numbered according to the silica content (molar ratio) e.g., gel of ZrO_2_:SiO_2_ = 1:2 was named 2T. etc., while samples obtained with zirconium (IV) acetate were signed as C series with the same numbering connected with SiO_2_ molar content. All gels were dried for four weeks in the air. The longest period of 28 days was required to dry the gels while maintaining the structure of samples without increasing the temperature. After drying, they were annealed with the speed of 4 °C/min up to 500 °C, 1000 °C and 1200 °C in air and kept for 30 min at each temperature. Temperatures of annealing were selected on the basis of DTA and TG measurements (not presented in this work). According to DTA and TG thermograms of a sample of ZrO_2_:SiO_2_ = 1:2 composition, as an example, the first endothermic effect was observed at about 89.5 °C, together with 18% weight loss and could be connected with the water and the solvent desorption. The exothermic effect was observed as the next one at 416.4 °C with 10% weight loss. This effect was assigned to the combustion and the decomposition of the organic components of the sample. In the thermograms of samples of higher ZrO_2_ content, two additional exothermic peaks were distinguished at about 536 °C and 850 °C which could be connected with the crystallization of the tetragonal ZrO_2_ which was running in two stages probably.

As the main research methods, FTIR (Fourier Transform InfraRed) spectroscopy and X-Ray diffraction (XRD) were selected. IR spectra were collected in VERTEX 70v Bruker spectrometer (Bruker, Billerica, MA, USA), 128 scans were collected at the resolution of 4 cm^−1^ and within Middle Infrared Range (MIR) 4000–400 cm^−1^. Measurements were run using the KBr pellets technique. X-Ray diffraction patterns were obtained during the measurements in the X`Pert Pro Panalytical diffractometer (Panalytical, Almelo, The Netherlands) applying Cu tube radiation and the Bragg-Brentano focusing. All samples were prepared as powders pressed in special holders and then measured with the step of 0.008° [2θ] and time 20 s for each step for the whole range from 5° [2θ] to 90° [2θ]. All diffraction patterns were analysed applying HighScore Plus software (version 3.0d (3.0.4) produced by: PANalytical B.V., Almelo, The Netherlands) and PDF-2 Release 2004 database bought together with the XRD diffractometer system. Then scanning electron microscopy (SEM) for the imaging of gels surfaces was applied. SEM imaging was performed in Fei NOVA NANO SEM 200 microscope (FEI Europe Company, Eindhoven, The Netherlands) together with a Genesis XM X-ray microanalysis system (EDAX, Tilburg, The Netherlands). Back-scattered electrons (BSE), as well as low vacuum (60 Pa) modes with a thin C layer, sputtered prior to measurements were used. During SEM imaging accelerating voltage of 15 and 18 kV was used, whereas during EDS measurements 18 kV was applied.

## Figures and Tables

**Figure 1 gels-08-00724-f001:**
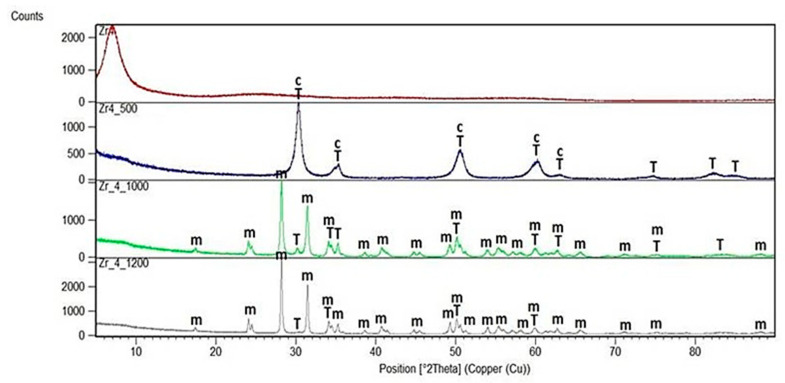
X-Ray diffraction pattern of pure ZrO_2_ gel (4% weight ZrO_2_) annealed at 500 °C, 1000 °C and 1200 °C in the air (m—monoclinic ZrO_2_, T—tetragonal ZrO_2_, c—cubic ZrO_2_).

**Figure 2 gels-08-00724-f002:**
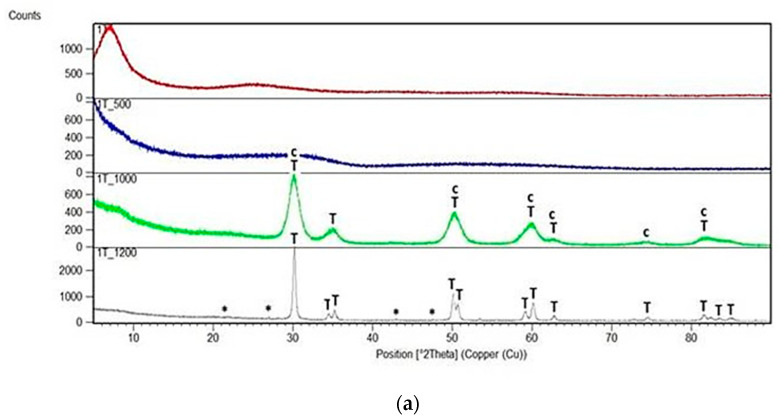
X-Ray diffraction patterns of: (**a**) 1T sample; (**b**) 2T sample; (**c**) 3T sample and (**d**) 4T sample. All samples were synthesized using zirconium (IV) n-propoxide and annealed at 500 °C, 1000 °C and 1200 °C in the air (*—undefined silica, T—tetragonal ZrO_2_, c—cubic ZrO_2_).

**Figure 3 gels-08-00724-f003:**
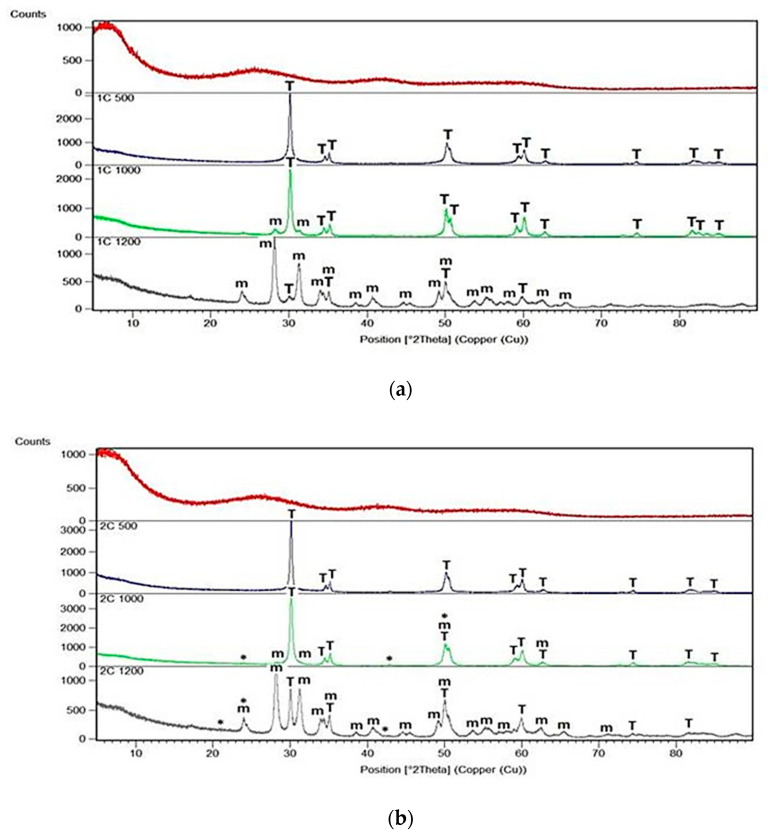
X-Ray diffraction pattern of: (**a**) 1C sample; (**b**) 2C sample; (**c**) 3C sample and (**d**) 4C sample. All samples were synthesized using zirconium (IV) acetate and annealed at 500 °C, 1000 °C and 1200 °C in the air (*—undefined silica, m—monoclinic ZrO_2_, T—tetragonal ZrO_2_, c—cubic ZrO_2_).

**Figure 4 gels-08-00724-f004:**
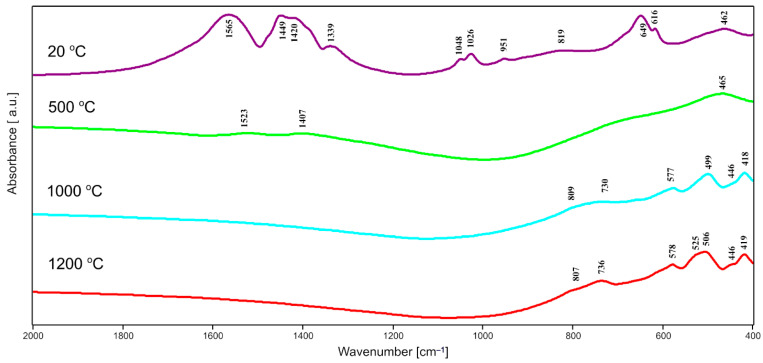
FTIR spectra of pure ZrO_2_ (4% mas. ZrO_2_), synthesized using zirconium (IV) n-propoxide and annealed at 500 °C, 1000 °C, and 1200 °C in air.

**Figure 5 gels-08-00724-f005:**
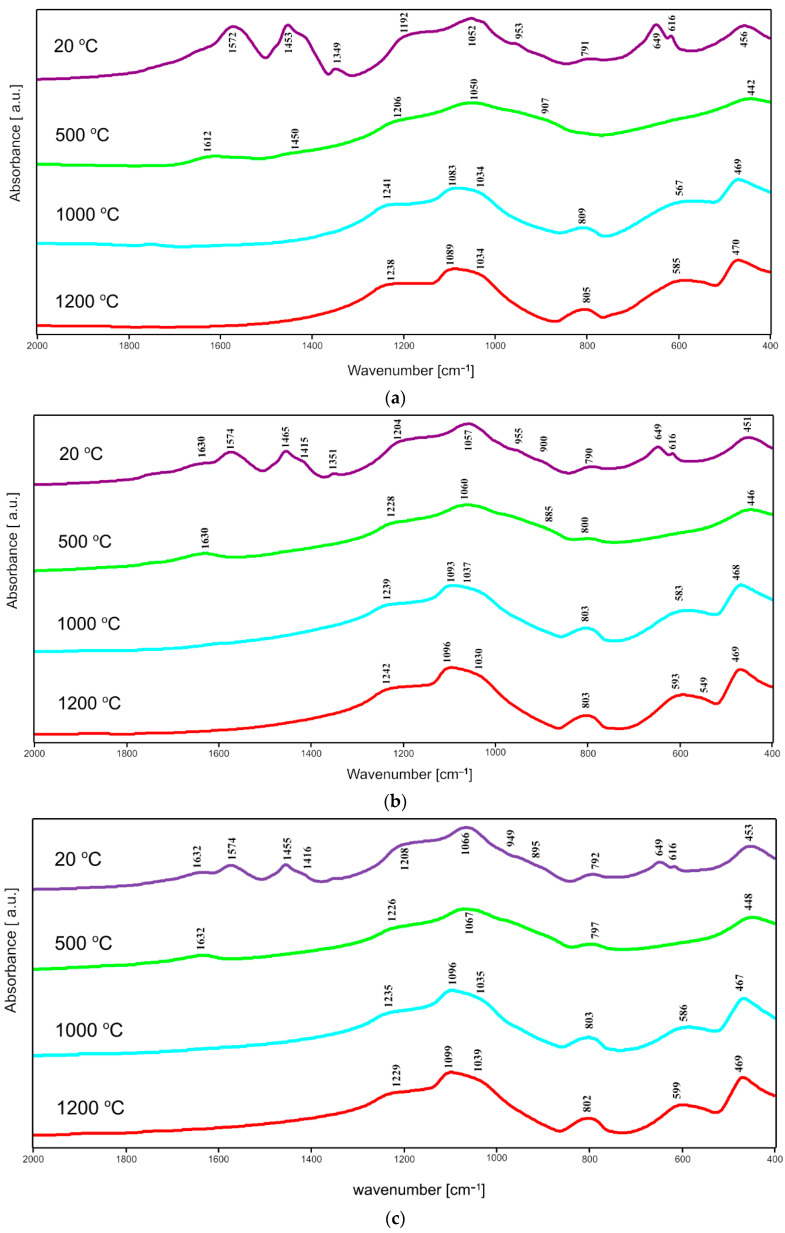
FTIR spectra of: (**a**) 1T sample; (**b**) 2T sample; (**c**) 3T sample and (**d**) 4T sample. All samples were synthesized using zirconium (IV) n-propoxide and annealed at 500 °C, 1000 °C and 1200 °C in the air.

**Figure 6 gels-08-00724-f006:**
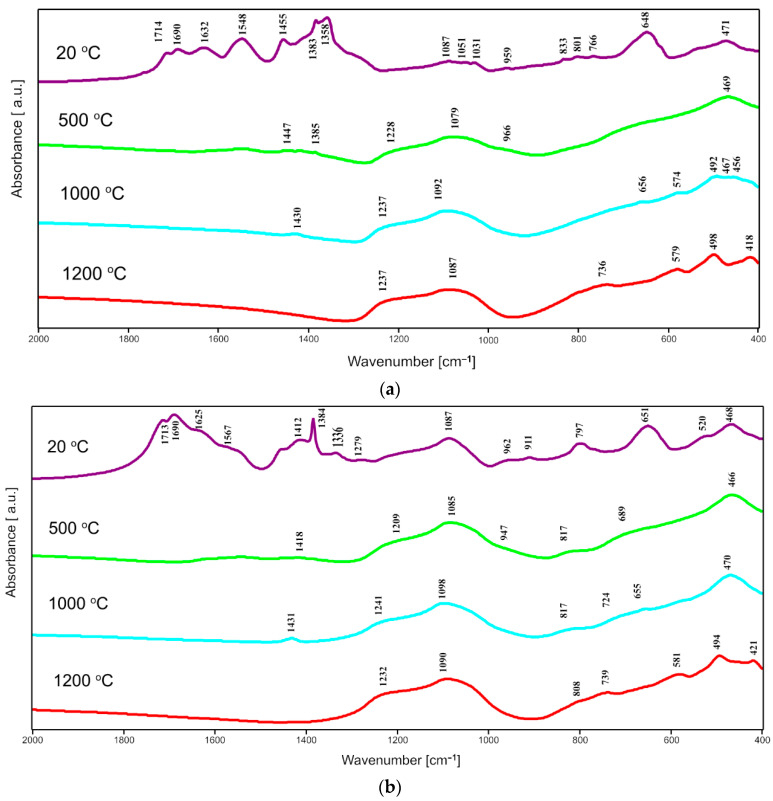
FTIR spectra of: (**a**) 1C sample; (**b**) 2C sample; (**c**) 3C sample and (**d**) 4C sample. All samples were synthesized using zirconium (IV) acetate and annealed at 500 °C, 1000 °C and 1200 °C in air.

**Figure 7 gels-08-00724-f007:**
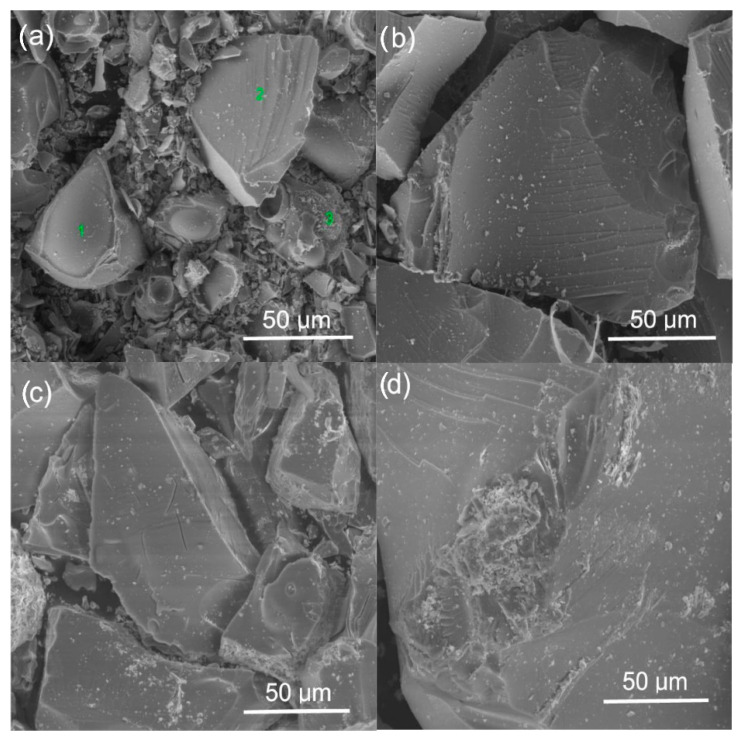
SEM images of: (**a**) 1T gel synthesized using zirconium (IV) n-propoxide; (**b**) 4T gel synthesized using zirconium (IV) n-propoxide; (**c**) 1C gel synthesized using zirconium (IV) acetate and (**d**) 4C gel synthesized using zirconium (IV) acetate. All gels were annealed at 1200 °C in air.

**Figure 8 gels-08-00724-f008:**
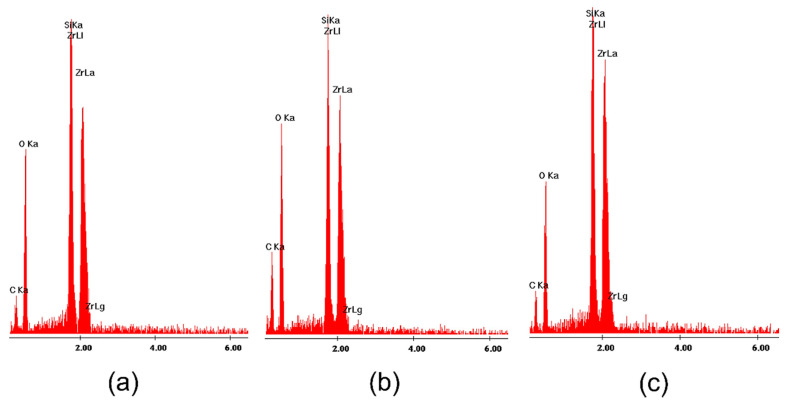
EDS analysis of 1T gel annealed at 1200 °C in the air (Figure 7a) at: (**a**) point 1; (**b**) at point 2 and (**c**) at point 3.

**Figure 9 gels-08-00724-f009:**
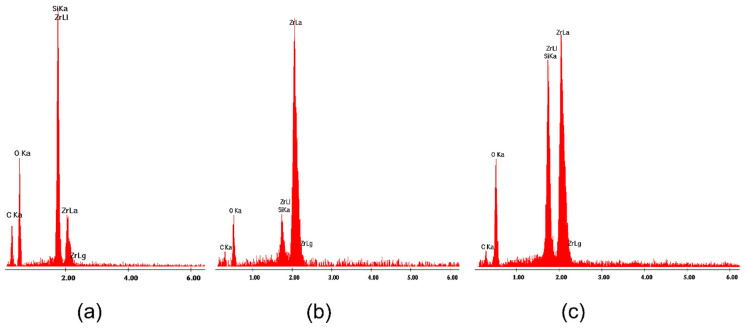
EDS analysis of the whole studied surface of gels: (**a**) 4T (Figure 7b); (**b**) 1C (Figure 7c) and (**c**) 4C (Figure 7d).

## Data Availability

Data is contained within the article.

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
