# Peer review of "The Study of the Influence of ZrO2 Precursor Type and the Temperature of Annealing on the Crystallization of the Tetragonal Polymorph of ZrO2 in Zirconia-Silica Gels"

_gels, 2022, doi:10.3390/gels8110724_

Round 1

Reviewer 1 Report

The paper focus on study the structure of the obtained materials in relation to ZrO2 precursor type applied during the synthesis.. The author also investigated the stabilization of the tetragonal polymorph of zirconia in materials by applying the different type precursor and the proper temperature of annealing.

 In general, the results mostly support the authors' conclusions. However, some aspects of the manuscript must be carefully reviewed, discussed and improved.

1°) The originality, mechanism, and scientific reliability of the work are clear. In my opinion, there are some major points that the authors should address before it is accepted for publication.

2°) Why do authors study the removal of this composite: ZrO2 precursor type? Authors must indicate why the use of this compound could be interesting. 

3°) please details more the part (give more information about) materials (MEB, BET, XRD analysis)

4°) line 100: leave a space between the numbers and the degree of the temperature (1000°C)

5°) Please add more description about “ the spherical crystallites of probably tetragonal polymorphs of ZrO2 in samples of series T of higher ZrO2 content (SEM)" in order to avoid any confusion

Author Response

Thank you very much for your work helping to improve my manuscript,

below there are answers for  your remarks:

Reviewer no 1 answers:

Ad. 2.  I hope I understand this question properly and the reviewer had in mind “ the use of ZrO2 “ not “the removal” ? I select ZrO2 and its precursors as a changing parameter in this two component system due to promising results which I described in two other articles ([27,26] positions in the refence list).  The focus was put on the possibility of obtaining tetragonal zirconia, known for its valuable properties, in the simplest possible way what was described in the introduction part.

Ad. 3. I added more details describing the parameters of XRD, FTIR and SEM (in English or MEB in French) methods in Materials chapter. I did not used BET in SEM measurements so I kindly ask for the explanation concerning this method.

In case of XRD method : “All samples were prepared as powders pressed in special holders and then measured with the step of 0.008o [2q] and time 20 seconds for each step for the whole range from 5o [2q] to 90o [2q]. All diffraction patterns were analysed applying HighScore Plus software and PDF-2 Release 2004 data base bought together with the XRD diffractometer system.“

In case of SEM method: “ Back-scattered electrons (BSE), as well as low vacuum (60 Pa) modes with thin C layer sputtered prior to measurements were used. During SEM imaging accelerating voltage of 15 and 18 kV was used, whereas during EDS measurements 18kV was applied.”

In case of FTIR: “128 scans were collected at the resolution of 4 cm-1 and within Middle Infrared Range (MIR) 4000-400 cm-1.”

Ad. 4. line 100: leave a space between the numbers and the degree of the temperature (1000°C): I made the correction at this place. I am not sure what the reviewer had in mind because 1000oC in line no 100 is written in the same way as in every other place  where the temperature value is given - without any spaces between number and degree  or between degree and letter C. I kindly ask for the more precision remark.

Ad. 5. I add more details concerning the spherical objects in SEM images of sample 1T:

“The same type spherical objects were observed on the surface of pure ZrO2 gel synthesized using zirconium (IV) n-propoxide and annealed up to 1200o C (data not presented in this work). The XRD measurement confirmed the presence of tetragonal zirconia in this samples. Similar crystallites were also reported in zirconia-silica samples annealed at 1200o C during the transformation from tetragonal to monoclinic polymorphs of ZrO2. Thus, one can attribute the spherical objects to crystallites of tetragonal zirconia which presence in this sample is also confirmed by XRD measurements.”

Reviewer 2 Report

This work demonstrates to achieve tetragonal phase of zirconia using two precursors with different ratio of silica and annealing temperatures in gel processing. It may suit the scope of this journal and would be of interest for the readers. However, the manuscript needs major revisions to be recommended for publication. Firstly, to rearrange the figures to fit in the pages and to show in better ways, that helps to follow and understand the description of the manuscript. Secondly to correct some discussions to harmonize with the obtained data.

Comments and questions are in the following.

1: P1, L12 and P16, L315           It is written that the gels are dried for 28 days.

Should the drying process be so long at RT or could it be at a higher temperature for a shorter time? I am curious what would be different in the obtained samples when drying at a temperature at which water and solvent desorbs, i. e. around 90 °C, as noted in the Material and Methods section, for example. 

2: P1, L18        “regular and tetragonal zirconia”

What does “regular” mean here? It is not clear to the readers what the author intends.

3: P2, L60-63      the sentence below might need correction. There seems something missing.   

Although one can find works where authors did not observe the correlation between the presence of t-ZrO2 and Zr precursor type as in case of applying a zirconium oxychloride, a zirconium oxynitrate and a zirconium acetate as Zr precursors and their influence on the gels structure [12].

4: The terms for the sample notation T and C

The author is using T and C for the two series, but this might confuse the readers with the crystalline symmetry, tetragonal and cubic, when tetragonal appear as "T" in XRD patterns. If the series are not correlated with the symmetry, I recommend to use other name for the series. Otherwise change the notation for the phase in XRD patterns. 

5: The sample series in page 5, L119-120 should be 3T and 4T instead of 3C and 4C?

6: P5, L 128    “with a very little contamination of silica” sounds a little strange.

Perhaps the term of “contamination” is not proper in this case, since it is natural that silica is there. The question is either it is observed in XRD or not, or if it is crystalline phase or not. Silica is there, so I recommend this to replace like, “with very small crystalline peaks of silica” or “with a very little amount of silica crystalline phase”.

7: FTIR spectra

       7-1: L144-147: It is written

In the FTIR spectra of all dried gels of pure ZrO2 and gels of series T and C (Figure 4-6) one can distinguish very similar bands at about 3500 cm-1 and at 1650 cm-1 assigned to the vibrations of OH- groups and the vibrations in the molecular water particles, respectively.

However, the band around 1650 cm-1 is observed only in some samples and is not very clear. It could be more obvious around 3500 cm-1, but one can hardly judge since the range is not shown.

7-2: P6, L169-170

 Though the author claims that the band at about 640-650 cm-1 can be found in the spectra of tetragonal form, the band is observed only in non-annealed samples. Perhaps small peaks may be notable in sample 1C and 2C, but then it does not match with the XRD results, since other samples containing more tetragonal ZrO2 phase do not show this band.

7-3: L177-180: the author claims that,

In the spectra of C series gels (Figure 6), the intensity of bands at 800 cm-1, 1100 cm-1 and 1220 cm-1 also increases but the mentioned earlier band at about 570-590 cm-1 appears only in spectra of samples 3C and 4C in which t-ZrO2 was identified, which agrees with XRD results.

However, sample 1C and 2C also show the tetragonal ZrO2 in XRD, which does not fit with this sentence, and the absorption band around 580 cm-1 are observed even more clearly than that of 3C and 4C in these two samples, 1C and 2C, of the highest temperature, for which monoclinic phase dominates. They are not consistent. 

7-4: L181-185:

     In the spectra of 1C and 2C samples a few weaker bands at 420 cm-1, 580 cm-1 and 730 cm-1 can be distinguished. Those bands can also be assigned to Zr-O connections but not only in the tetragonal form of ZrO2.

    That is why, according to the IR spectra, there is no possibility of the existence of continuous silica network in the studied zirconia-silica samples.

The existence of these three bands may be attributed to the different form of ZrO2 as is observed in XRD, i. e. monoclinic structure plus small amount of tetragonal phase. However, I do not understand why this result of IR spectra could explain there is no possibility of the existence of continuous silica network. From XRD, provided that the crystalline peaks are only of ZrO2 phases, it is clear that there is no continuous silica network, i. e. periodic structure of silica, since there is no such diffraction peak observed.   

8: P11- Figure 7-10

     The EDS spectra may better be omitted in the manuscript. Instead of the EDS spectra, list of elemental ratios of the points may be enough and even more informative in this case. It is also not good when the parts of one figure appears in many pages. I recommend to make 1 figure including the four SEM pictures from fig. 7-10 plus a table of the elemental ratio. Since the paper will be viewed on PCs, the SEM pictures could be more compact in the manuscript, so that they appear together in a page. In addition, please provide the SEM images in the same magnification for those four, if possible. Scale bar should also be clear in the picture.

Anyway, there is one critical question. The EDS spectra show that the zirconia and silica ratio between T and C series is very different after annealing. The silica ratio in C series seems much less compared to that of T series. The reason could be either silica is lost during the processing or there are silica rich parts in the sample other than the point measured for EDS. In any case, an apparent difference between T and C series is the silica ratio in the obtained grains. If we consider the influence of silica ratio in the grains, 1T might corresponds to 4C, of which XRD patterns and IR spectra of 1200°C are very similar indeed. It is quite crucial and very interesting, but such a point is not discussed in the manuscript.  

Considering the XRD, IR and EDS results, it seems the story may be rather simple. T series includes as much silica matrix in the grains so that the silica limits the zirconia crystals to grow, which results in more cubic phase especially at higher content of silica. In the case of C series, silica content in the grain is reduced so that zirconia grows similarly like the sample without silica to result in monoclinic phase when starting silica ratio is lower. The key issue would be the role of precursor to prevent silica matrix to be included in the grains like in the C series.   

9: P16, L273-274, “SEM studies allowed to observe the spherical crystallites of probably tetragonal polymorphs of ZrO2 in samples of series T of higher ZrO2 content.

I am not able to agree with this observation of the SEM picture. The SEM pictures in this manuscript would not help much to understand the observation, especially when we compare them with C series.

10: English text: in general, the English in the manuscript is fine. But I recommend the author to check the text carefully to revise some sentences and correct grammatic errors. Here is an example.  

P16-17, L266-269, could be revised as

In samples of series C synthesized applying zirconium acetate, the first detected phase was tetragonal t-ZrO2, which converted into monoclinic m-ZrO2 with annealing temperature in sample 1C and 2C (rich in ZrO2), but t-ZrO2 remained as the predominant phase in samples 3C and 4C (lower ZrO2 ratio) even at the highest annealing temperature.

Author Response

Thank you for all remarks which help to improve my manuscript.

Reviewer no 2 answers:

Ad. 1. I was trying to find out the simplest way of obtaining the tetragonal zirconia that is why I select slow drying of gels under ambient conditions. Period of 28 days is a time for all gels to be dried however those of higher zirconia content were drying quicker. Besides, there is a possibility that rising the temperature of drying up to 90o C might influence on the structure of samples during the further annealing. That is why I would not like to introduce another parameter during the structure research.

Ad. 2.  Excuse me for this mistake. I had in mind “cubic” nor “regular” zirconia. I will correct this phrase certainly.

Ad. 3. This sentence is really too long and one phrase was missed. I changed this and created two sentences instead:

“Although one can find works where authors did not observe the correlation between the presence of t-ZrO2 and Zr precursor type. Such situation was observed in case of applying three zirconia precursors: a zirconium oxychloride, a zirconium oxynitrate and a zirconium acetate, their lack of influence on the gels structure and the crystallization of tetragonal ZrO2 was described in [12]”.

Ad. 4. The series of samples were named as T and C according to zirconia precursors applied. Those names are used through the whole manuscript and also appear in particulate Figures (also at each diffraction pattern) so it is rather impossible to change them for other names. The XRD analysis is the only place where T and C letters wereduplicated as names but they can be easily distinguish when they mean zirconia polymorphs or when they are names of the samples (see Figures 1,2, 3 and their captions, please). In my opinion the introducing of other symbols for zirconia polymorphs than “c” for cubic structures and “t” for tetragonal ones will create the greater confusion than leaving the version of description of samples and polymorphs as applied in this article.

Ad. 5. Yes, there should be of course 3T  and 4T (instead 3C and 4C) because in this paragraph the diffraction patterns of gels of T series were described. It would be corrected.

Ad. 6. The use of word “ contamination” was used to point on a very little amount of a crystalline silica  but of course a silica is not a real contamination here but a component. The suggested phrase: “with a very little amount of silica crystalline phase” is much better and I replaced the phrase containing “contamination” with it.

Ad. 7-1. All spectra were measured in the range of 4000 cm-1 – 400 cm-1. There were two opportunities of the spectra presentation: in whole range or in the selected range. I chose the second one to make all, even weak bands, being visible in Figures. Both bands, this wide one at about 3500 cm-1 (behind the presented range) and the second one at 1650 cm-1 are not observed after annealing over 500o C what confirms the lack of OH- groups in the gels structure. It means that the mentioned bands should not influence on the crystallization of t-ZrO2. Thus I did not pay more attention to the mentioned bands.

Ad. 7-2. I claimed, according the reference [7] that the band at about 650 cm-1 can be due to the stretching vibrations of Zr-O bond in tetragonal zirconia, especially when the presence of this polymorph was confirmed by the XRD analysis. This possibility does not exclude the assignment of the mentioned band to Zr-O vibrations in other zirconia polymorphs. So this band cannot be treated as a kind of marker of t-ZrO2 but can be connected with this phase when t-ZrO2  presence is confirmed by other methods. Moreover, the bands due to the vibrations of Si-O bonds can also be located in the range of 560 cm-1 – 650 cm-1 that is why bands in this range can be observed in all samples containing Si-O and Zr-O bonds. By compering data obtained from two (e.g. FTIR and XRD) or three research methods one can try to assign bands in the mentioned range to proper bonds.

Ad. 7-3. Very weak bands at about 600-650 cm-1 can be observed in the FTIR spectra of gels 3C and 4C. But probably those bands were shifted to the positions 560 – 580 cm-1 after annealing – this can be observed in most samples not only belonging to C series.  The wide band at 580 cm-1 can also be a superposition of bands due to the Zr-O bond and Si-O bond vibrations, what can be confirmed by the increasing intensity of this band at 580 cm-1 in correlation of the rising SiO2  content. I agree that weak bands at about 650 cm-1 can be also observed in the spectra of 1C and 2C gels – this part will be corrected – but simultaneously one can connect this band with the tetragonal zirconia if this phase presence is confirmed by XRD analysis or in other case, this band in the range of 580 - 650 cm-1 can be assigned to the vibrations of Zr-O linkage in other ZrO2 polymorphs.

Ad. 7-4. Certainly, basing on the XRD analysis, one can confirm that the crystalline silica does not exist in samples (or can be identified as a phase of very little concentration) but the complete data set is obtained using both methods, the XRD diffraction and the FTIR spectroscopy which are an excellent complementary structure research methods devoted to the far order structures and allow to examine first coordination range, respectively.  That is why I applied both methods to draw the conclusions on the structure of studied materials.

Ad. 8. While preparing Figures containing SEM images, MDPI GELS template, available in Instructions for Authors was applied. According to the instructions, if Figure consists of a few images, they should appear one after the other, as a), then b)  etc. All SEM images presented in the manuscript are of the same 1000x magnification, so they can be compared in easy way. The parameters of SEM imaging as e.g. the accelerating voltage, a type of detector, a scale are legible below the each image.

Regarding the EDS spectra, I prefer to present a chemical compositions of the whole scans or at the selected points in this way to compare in more clear way the relation between Zr, Si or O concentrations at studied places than to compare the absolute values of elemental ratios at these points. Besides, these values can differ for the closely located points because of the radius of an beam applied for the analysis.

I agree with the conclusion that silica ratio differs in case of T and C series what can be connected with the distribution of silica in the samples structure. The EDS or suggested elemental ratios are very local measurements depending on the selected area or a point of measurement what confirms that observed results can be connected with the changes of the local distribution of silica. All this confirms that there are areas rich of silica (first of all as amorphous phase, with a little addition of crystalline silica in case of some samples)  - they may be called: grains and parts of samples of higher zirconia content where the crystalline polymorphs of ZrO2 are observed. Thus there is a conclusion that samples structure is built of silica matrix (an amorphous and/or crystalline) with crystalline zirconia phases incorporated into it. The influence of zirconia precursor type on gels structure changes is indisputable and obvious.

Ad. 9. As it was mentioned in the manuscript, SEM images of gels of series T and C dried and annealed do not differ in a distinct way from each other with one exception of sample 1T where the spherical objects were observed. It was suggested that these objects can be connected with a tetragonal zirconia phase. Such crystallites were also observed in SEM images of pure zirconia gel after annealing at 1200o C (not presented in this work) where the presence of t-ZrO2 together with m-Zro2 was confirmed by the XRD analysis. Such shapes of t-zirconia crystallites were also noticed by the authors of reference [10] in zirconia-silica gels. I added more information on these objects in part concerning SEM.

Ad. 10. Thank you for this remark. I substituted “conversed” by “converted”.

Round 2

Reviewer 2 Report

I can acknowledge the corrections of the author according to my comments, but there are issues which I do not agree in the response of the author. Please reconsider the points to improve the quality of this paper. I am suggesting them which are crucial for publication. In the following, my comments are in blue texts.

Ad. 1. I was trying to find out the simplest way of obtaining the tetragonal zirconia that is why I select slow drying of gels under ambient conditions. Period of 28 days is a time for all gels to be dried however those of higher zirconia content were drying quicker. Besides, there is a possibility that rising the temperature of drying up to 90o C might influence on the structure of samples during the further annealing. That is why I would not like to introduce another parameter during the structure research.

My point is indeed these arguments, which is missing in the manuscript. Please include this in a short sentence that longest period of 28 days is required to dry the gels while maintaining the structure of samples without increasing the temperature.

Ad. 3. This sentence is really too long and one phrase was missed. I changed this and created two sentences instead:

Although one can find works where authors did not observe the correlation between the presence of t-ZrO2 and Zr precursor type. Such situation was observed in case of applying three zirconia precursors: a zirconium oxychloride, a zirconium oxynitrate and a zirconium acetate, their lack of influence on the gels structure and the crystallization of tetragonal ZrO2 was described in [12]”.

The first sentence is grammatically incorrect. “Although” should be replaced with “However”.

Ad. 7-1. All spectra were measured in the range of 4000 cm-1 – 400 cm-1. There were two opportunities of the spectra presentation: in whole range or in the selected range. I chose the second one to make all, even weak bands, being visible in Figures. Both bands, this wide one at about 3500 cm-1 (behind the presented range) and the second one at 1650 cm-1 are not observed after annealing over 500o C what confirms the lack of OH- groups in the gels structure. It means that the mentioned bands should not influence on the crystallization of t-ZrO2. Thus I did not pay more attention to the mentioned bands.

From this explanation, one could understand better. In this case, the sentence in the manuscript L 146-149, “In the FTIR spectra of all dried gels of pure ZrO2 and gels of series T and C (Figure 4-6) one can distinguish very similar bands at about 3500 cm-1 and at 1650 cm-1 assigned to the vibrations of OH- groups and the vibrations in the molecular water particles, respectively.” should be revised. One can NOT distinguish in the plots if there are these peaks at 3500 cm-1 since these regions are not shown. If you would not think it is not necessary to show the region, then omit (Figure 4-6) and add this in the later when it comes to the discussion. I would suggest writing the sentence in the following.  

In the FTIR spectra of all dried gels of pure ZrO2 and gels of series T and C, corresponding bands are observed at about 3500 cm-1 and at 1650 cm-1 assigned to the vibrations of OH- groups and the vibrations in the molecular water particles, respectively. In figure 4-6, one can distinguish the band at 1650 cm-1 up to 500 °C.

Ad. 7-2. I claimed, according the reference [7] that the band at about 650 cm-1 can be due to the stretching vibrations of Zr-O bond in tetragonal zirconia, especially when the presence of this polymorph was confirmed by the XRD analysis. This possibility does not exclude the assignment of the mentioned band to Zr-O vibrations in other zirconia polymorphs. So this band cannot be treated as a kind of marker of t-ZrO2 but can be connected with this phase when t-ZrO2  presence is confirmed by other methods. Moreover, the bands due to the vibrations of Si-O bonds can also be located in the range of 560 cm-1 – 650 cm-1 that is why bands in this range can be observed in all samples containing Si-O and Zr-O bonds. By compering data obtained from two (e.g. FTIR and XRD) or three research methods one can try to assign bands in the mentioned range to proper bonds.

This explanation is not consistent with the data shown in this manuscript. My question is „why the band around 650 cm-1 is NOT visible in annealed samples 1T, 2T and 3C, 4C of which XRD patterns clearly show tetragonal phases?”. When it is not visible in the data given in the manuscript, why one can discuss about this band? One can claim that it could be visible according to the literatures, but cannot discuss in this way when it is not visible. It does not deny the XRD results, and I believe that they do have tetragonal phases. It does not make sense to discuss further when one cannot observe them in FTIR.

Moreover, I could not find the explanation in the reference [7] that this band is attributed to the band “in tetragonal zirconia”. If this is wrong, please show where it is presented.

Ad. 7-3. Very weak bands at about 600-650 cm-1 can be observed in the FTIR spectra of gels 3C and 4C. But probably those bands were shifted to the positions 560 – 580 cm-1 after annealing – this can be observed in most samples not only belonging to C series.  The wide band at 580 cm-1 can also be a superposition of bands due to the Zr-O bond and Si-O bond vibrations, what can be confirmed by the increasing intensity of this band at 580 cm-1 in correlation of the rising SiO2  content. I agree that weak bands at about 650 cm-1 can be also observed in the spectra of 1C and 2C gels – this part will be corrected – but simultaneously one can connect this band with the tetragonal zirconia if this phase presence is confirmed by XRD analysis or in other case, this band in the range of 580 - 650 cm-1 can be assigned to the vibrations of Zr-O linkage in other ZrO2 polymorphs.

When the bands at about 600-650 and 560-580 are observed in the same sample 1C simultaneously, as the author indicated in the graph, then the latter could not be attributed to the same band of the former which is shifted. This is not rational. The same way like in my feedback to Ad. 7-2, it is not necessary to claim that these bands are attributed to the tetragonal zirconia from FTIR. XRD results already show that and FTIR is not very obvious. It is possible but no one could claim that from this. I do not find a reason to claim like this here.

Ad. 7-4. Certainly, basing on the XRD analysis, one can confirm that the crystalline silica does not exist in samples (or can be identified as a phase of very little concentration) but the complete data set is obtained using both methods, the XRD diffraction and the FTIR spectroscopy which are an excellent complementary structure research methods devoted to the far order structures and allow to examine first coordination range, respectively.  That is why I applied both methods to draw the conclusions on the structure of studied materials.

My question was, “I do not understand why this result of IR spectra could explain there is no possibility of the existence of continuous silica network. From XRD, provided that the crystalline peaks are only of ZrO2 phases, it is clear that there is no continuous silica network, i. e. periodic structure of silica, since there is no such diffraction peak observed. „ It is not an answer to this question.

I agree that the combination of XRD and IR studies could complement each other. However, compared to XRD, in which longer range order appears clearly, IR data here would not show evidence of such an inexistence of continuous network. Therefore, it is not understandable that the author summarizes from IR,

Summarizing, according to the IR spectra (together with the results of the XRD analysis) there is no possibility of the existence of continuous, periodic silica network in the studied zirconia-silica samples.“ (L186-188)

Ad. 8. While preparing Figures containing SEM images, MDPI GELS template, available in Instructions for Authors was applied. According to the instructions, if Figure consists of a few images, they should appear one after the other, as a), then b)  etc. All SEM images presented in the manuscript are of the same 1000x magnification, so they can be compared in easy way. The parameters of SEM imaging as e.g. the accelerating voltage, a type of detector, a scale are legible below the each image.

There are many examples in this journal which show the pictures in reasonable ways. Please have a look of these to improve the appearance of the data. Here are the examples from the recent publication in this journal. The letters in the SEM images (figure 7-10) are too small and not clear. The parameters of SEM could be given separately, if necessary, but the scale is to be clear like those in the following papers.

Fig. 1 and 2 in https://mdpi-res.com/d_attachment/gels/gels-08-00691/article_deploy/gels-08-00691.pdf?version=1666780099

Fig. 8 in https://mdpi-res.com/d_attachment/gels/gels-08-00675/article_deploy/gels-08-00675-v2.pdf?version=1666347444

Fig. 3 in https://mdpi-res.com/d_attachment/gels/gels-08-00634/article_deploy/gels-08-00634-v2.pdf?version=1665213504

Regarding the EDS spectra, I prefer to present a chemical compositions of the whole scans or at the selected points in this way to compare in more clear way the relation between Zr, Si or O concentrations at studied places than to compare the absolute values of elemental ratios at these points. Besides, these values can differ for the closely located points because of the radius of an beam applied for the analysis.

I agree with the conclusion that silica ratio differs in case of T and C series what can be connected with the distribution of silica in the samples structure. The EDS or suggested elemental ratios are very local measurements depending on the selected area or a point of measurement what confirms that observed results can be connected with the changes of the local distribution of silica. All this confirms that there are areas rich of silica (first of all as amorphous phase, with a little addition of crystalline silica in case of some samples)  - they may be called: grains and parts of samples of higher zirconia content where the crystalline polymorphs of ZrO2 are observed. Thus there is a conclusion that samples structure is built of silica matrix (an amorphous and/or crystalline) with crystalline zirconia phases incorporated into it. The influence of zirconia precursor type on gels structure changes is indisputable and obvious.

It is more important that the results are clearly explained in either case: the EDS results are average of whole specimen or in the local area. If all the grains in the same sample show significant differences in elemental ratio, why such spectra are shown? It is more important to know what the major difference in the elemental ratio and microstructure between the samples is, and what is the difference in obtained structure, and to see what is correlated between them. When the author claims that the elemental ratio is only the local information so that one cannot compare among the different samples, it does not make sense to show the EDS results. If it is true, that the grains are only the part which has different silica ratio in the case of C series, then please show the evidence that the other parts (not shown by EDS in the manuscript) in series C has more silica so that one can prove the consistency of the elemental ratio similar to that of the starting mixture.

I would recommend to reconsider this point and if the author thinks the EDS data shown in the manuscript is more or less reflecting the characteristic of each sample, it is valuable to discuss in the manuscript. If the author would like to show the EDS spectra, I also recommend having a table like in the following paper, which is also from this journal. I stress here that the space in the paper should be used in an efficient way and more important is to display in a reasonable way so that it is easy to read and understand. This is not a thesis but a journal paper.

Fig. 1 and Table 1 in https://mdpi-res.com/d_attachment/gels/gels-08-00690/article_deploy/gels-08-00690.pdf?version=1666778723

Author Response

Thank you once more for your hard work to improve my article.

Reviewer`s remarks responses:

Ad. 1. I added the suggested sentence into the manuscript , in Materials section. I did not add this sentence into the abstract to avoid making the abstract being longer.

“Longest period of 28 days is required to dry the gels while maintaining the structure of samples without increasing the temperature”.

Ad. 3. I replaced “ Although” by “ However”  as it was suggested.

Ad. 7-1. I introduced the suggested sentence: “ In the FTIR spectra of all dried gels of pure ZrO2 and gels of series T and C, corresponding bands are observed at about 3500 cm-1 and at 1650 cm-1 assigned to the vibrations of OH- groups and the vibrations in the molecular water particles, respectively. In Figures 4-6, one can distinguish the band at 1650 cm-1 up to 500 °C.” into the manuscript  instead of my version of the mentioned part.

Ad. 7-2. The band at about 610-650 cm-1 does not appears in the spectra of samples of higher zirconia content (gels 1T and 2T)  synthesized with zirconium (IV) n-propoxide and in spectrum of samples 3C and 4C (of higher silica content) obtained with Zr acetate.  This band is rather a weak one and its absence in the mentioned spectra can be connected with a few reasons. It can be the dependence between the zirconia precursor type and silica content. As it was mentioned before, in the range of 400 – 700 cm-1 one can find another bands due to Si-O vibrations. The low concentration of Zr-O bonds in the structure and higher concentration of Si-O bonds can also influence on the visibility of the discussed band – although silica is mostly in an amorphous state or poorly crystallized. I think this band is worth to be mentioned (although not visible in all spectra) because it can be connected with the presence of t-zirconia.

What is concerning data given in [7] reference. The band at about 600 – 650 cm-1 is assigned to the vibration of ZrO8 group (see Table VI).  Such structural group points on 8 as the coordination number of Zr. So Zr is surrounded by 8 oxygens which create the cube as the coordination polyhedron. Such coordination is typical for the tetragonal form of zirconia [see e.g.  D. K. Smith, H. W. Newkirk, Acta Cryst., 18 (1963) 983-991; G. Teufer, Acta Cryst., 15 (1962) 1187). In monoclinic form of zirconia, the coordination number of Zr equals 7 so it is easy to distinguish both Zr containing structural groups in tetragonal and/or monoclinic polymorphs. Thus when the band at 650 cm-1 is assigned to ZrO8 group as is given in [7], one can treat it as assigned to Zr-O bond vibrations in the tetragonal zirconia.

Ad. 7-3.  The presence of both bands at about 650 cm-1 and 560 cm-1 in the spectrum of 1C gel annealed at 1000oC is the only exception when both bands are observed together. The band at 650 cm-1 in this spectrum is very weak and it can be possible that it originates from other vibrations that that of Zr-O bond (e.g. from the pseudolattice vibrations of structure built up of silica rings). In all other spectra of annealed gels one can observe only bands at about 560 cm-1 or only those at about 580 cm-1. So there is still possible in samples spectra that band at 650 cm-1 is shifted to 560 cm-1 position after annealing of gels what I have suggested.

Ad. 7-4. One can observe the changes in shape and intensity of typical of Si-O vibrations bands e.g. at about 1100 cm-1 assigned to the asymmetric stretching vibration of Si-O bond , at about 1220 cm-1 assigned to the double Si=O bond vibrations and at 940 cm-1 due to the vibrations of Si-O- broken bridges. The last two bands are observed in selected spectra. When the intensity of band at 1100 cm-1 increases it points on the progressive polymerization in the silica lattice, But when at the same time, the intensity of the band at 1220 cm-1 also increases it indicates the parallel depolymerization of silica structure. Basing on the observation of the changes of bands characteristic for Si-O bonds vibrations, one can draw conclusions on the silica matrix structure. The XRD analysis gives the complementary structural data. Both methods allow to obtain “the complete picture” of the samples structure, what was mentioned (as you noticed) „Summarizing, according to the IR spectra (together with the results of the XRD analysis).

Ad. 8. I prepared new Figures presenting SEM and EDS results (Figures 7, 8 and 9). I hope they will be acceptable in this version. Figures 8 and 9 allow to compare the results of EDS analysis in a simple way so I decided not to add the Table. As I mentioned earlier, the results of EDS analysis, without using the standards are connected with rather big mistakes so I used them more to a qualitative analysis than to exactly to quantitative one.

The problem with the disturbances of the relation Zr:Si in gels synthesized with Zr acetate (series C) can be also connected with a bigger tendency of zirconia phases crystallization in these gels what also could be connected with the quicker hydrolysis and polycondensation of sols of this series. One could observe during the polycondensation of gels that all process begun on the surface. During the annealing, the crystallization of zirconia polymorphs was continued what could influence on the Zr concentration changes in the studied samples.

I mentioned this in the manuscript:

“Although the surface of the studied samples look very similar (with this mentioned exception of gel 1T), analysing EDS spectra one can observe the disturbances of the relation of Zr : Si atoms concentration in gels of series C synthesized with Zr (IV) acetate (Figure 9) with respect to the planned compositions of samples. It might be connected with the local fluctuation of both Zr and Si atoms concentration in samples which may result from the synthesis conditions and also might influence on a faster crystallization of zirconia polymorphs during the annealing of samples”.